# Magniber v2 Ransomware Decryption: Exploiting the Vulnerability of a Self-Developed Pseudo Random Number Generator

**Sehoon Lee [1], Myungseo Park [1,*] and Jongsung Kim [1,2]**

1   Department of Financial Information Security, Kookmin University, Seoul 02707, Korea;
    dreamtree304@kookmin.ac.kr (S.L.); jskim@kookmin.ac.kr (J.K.)
2   Department of Information Security, Cryptology and Mathematics, Kookmin University, Seoul 02707, Korea
*   Correspondence: pms91@kookmin.ac.kr

**Abstract:** With the rapid increase in computer storage capabilities, user data has become increasingly important. Although user data can be maintained by various protection techniques, its safety has been threatened by the advent of ransomware, defined as malware that encrypts user data, such as documents, photographs and videos, and demands money to victims in exchange for data recovery. Ransomware-infected files can be recovered only by obtaining the encryption key used to encrypt the files. However, the encryption key is derived using a Pseudo Random Number Generator (PRNG) and is recoverable only by the attacker. For this reason, the encryption keys of malware are known to be difficult to obtain. In this paper, we analyzed Magniber v2, which has exerted a large impact in the Asian region. We revealed the operation process of Magniber v2 including PRNG and file encryption algorithms. In our analysis, we found a vulnerability in the PRNG of Magniber v2 developed by the attacker. We exploited this vulnerability to successfully recover the encryption keys, which was by verified the result in padding verification and statistical randomness tests. To our knowledge, we report the first recovery result of Magniber v2-infected files.

**Keywords:** ransomware; magniber; decryption; cryptography

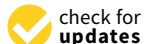

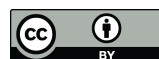

## 1. Introduction

### 1.1. Background

Ransomware, a compound word of ransom and software, is a malicious software that encrypts important files such as documents, photos and videos and requires payment for data recovery. Enabled by advanced IT technology, increased PC penetration rate, and the advent of Bitcoin that allows anonymous transactions, ransomware attacks can accrue financial benefits. Ransomware has transformed from non-targeted infection of unspecified targets to targeted infection of companies or institutions. In March of 2019, the Norwegian aluminum producer Norsk Hydro was infected with LockerGoga, a ransomware that enforced the shutdown of the extrusion process, and the separation of all the company's factories and operations networks from the global network. During this process, some automated processes were switched to manually operation. The known damage was approximately US $41 million [1]. In early March of 2019, computers in the government offices of Jackson County, Georgia, were infected with Ryuk and many jobs except 911 were paralyzed. As Jackson County City's backup systems were also encrypted, the government paid 100 bitcoins for quick recovery [2]. If the required ransom is paid, the attacker notices not only the victims of the individual or group victims, but also that their industry is willing to pay for recovering the information. It was also recently reported that ISS World, which provides cleaning, catering, security and other services in UK, has become a victim of a ransomware in 2020. The company's website has been down since Feb and the management at the London's Surrey, Canary Wharf and Weybridge offices

consisting of 43,000 staff members are inaccessible emails since then. ISS officials say that the database has been locked from being accessed because of file-encrypting malware infection [3]. To overcome the damage caused by ransomware, a fundamental solution other than ransom payment is required. One effective solution is recovery of the encryption key that encrypted the file. Ransomware generates encryption keys by a PRNG. After encrypting the files in the victim's PC, the encryption keys are protected and can be recovered only by the attacker. One protection method is hybrid encryption that combines symmetric- and asymmetric-key ciphers. The high-speed symmetric-key cipher is used for data encryption and the asymmetric-key cipher is used for encrypting the encryption key of the symmetric-key cipher. File encryption is performed by a symmetric-key cipher: the attacker encrypts the encryption keys using their public key and places it in the victim's system. The encryption keys are safely protected because only the attacker's private key, which is not stored in the victim's system, is imperative information for decrypting them. Therefore, obtaining the encryption keys is notoriously difficult.

　　　Magniber, a variant of Cerber ransomware, is mainly targeted South Korean victims in the mid-2017. According to AhnLab statistics, Magniber's sample rate was highest from the fourth quarter of 2017 to the first quarter of 2018 [4]. The two existing versions of Magniber (versions 1 and 2) were distributed via the Magnitude exploit kit. A main feature of Magniber v1 is the encryption of all files by the same encryption key, which is received from the Command and Control (C&C) server. If the C&C server closes or the network is disconnected, the encryption key is not received and a hard-coded value is used instead [5]. The hard-coded encryption key is identified by inspecting the extension of the infected file. The decryption tool distributed by AhnLab can decrypt Magniber v1-infected files using the hard-coded value, but cannot yet decrypt the encryption key received from the C&C server [6]. Magniber v2 uses a hybrid encryption that combines the symmetric-key cipher AES and the asymmetric-key cipher RSA. All files infected by Magniber v2 are encrypted with AES using different encryption keys, which are themselves encrypted using the attacker's RSA public key. In other words, the victims need to obtain the attacker's RSA private key to decrypt the encrypted AES encryption keys and hence recover the infected file. However, the asymmetric-key cryptography ensures that the attacker's private key is only owned by the attacker and cannot be obtained by another party. In addition, cryptographic algorithms (including AES and RSA used in Magniber v2) have been continuously researched for decades, ensuring sufficient security. For these reasons, Magniber v2 remains as unrecoverable ransomware to date. Table 1 summarizes the features of Magniber v1 and v2.

**Table 1.** Comparison of Magniber version 1 and 2.

| Ransomware | Magniber v1 | Magniber v2 |
| --- | --- | --- |
| Crypto system | AES128-CBC-PKCS7Padding | AES128-CBC-PKCS7Padding<br>RSA2048-OAEP |
| Key generation | 1. Receive from C&C server<br>2. Fixed in code | Attacker's PRNG |
| Key management | None | Encrypted AES key with RSA<br>stored in end of file |
| Key destruction | Yes | Yes |
| Decryption tool | Yes | No |
| Known extensions | kympzmzw, owxpzylj, prueitfik, rwighmoz, bnxzoucsx, tzdbkjry, iuoqetgb, pgvuuryti, zpnjelt, gnhnzhu, hssjfbd, ldolfoxwu, zskgavp, gwinpyizt, hxzrvhh, cmjedin, dzvtwtqz, pxynindl, sqzprtt, etc. | fprgbk, ihsdj, kgpvwnr, vbdrj, skvtb, vpgvlkb, dlenggrl, dxjay, fbuvkngy, xhspythxn, demffue, mftzmxqo, qmdjtc, wmfxdqz, ndpyhss, dyaaghemy, etc. |

　　　In this paper, we provide (to our knowledge) the first decryption of files infected by Magniber v2. Magniber v2 generates encryption keys for file encryption using a PRNG developed by the attacker. However, our analysis revealed that Magniber's PRNG is not cryptographically secure. This crucial weakness is essential for encryption keys recovery.

We were able to reduce the range of encryption key candidates by exploiting the vulnerability of the PRNG. The encryption key was then recovered by verifying whether the ciphertext decrypted by the encryption key candidates is plaintext or not. To distinguish between plaintext and ciphertext, we employed padding verification and the statistical randomness tests of the National Institute of Standards and Technology (NIST) SP800-22 [7]. For efficient encryption key recovery, we first performed a padding verification, which has a relatively low execution cost, then the second verification through statistical randomness tests on the encryption key candidates that passed the first verification. The correct encryption key was then selected as the encryption key candidate that passed both verifications. Using this key, we were able to decrypt the infected files.

*1.2. Our Contributions*

This paper analyzes the encryption process of Magniber v2 and proposes a method that decrypts Magniber v2 without requiring the attacker's private key. Our contributions are summarized below.

1.  The existing decryption technique for Magniber-infected files decrypts only version 1, in which the main parameters are hard-coded key. Files infected by Magniber v2, which uses hybrid encryption, cannot be decrypted. We present the first successful decryption of Magniber v2-infected files. Our proposed method is experimentally verified in a demonstration.
2.  The existing ransomware generates the encryption keys for file encryption by a random number generator (RNG). A typical RNG determines the security strength from seeds generated by collecting various noise sources. Well-known RNGs such as Hash DRBG and CTR DRBG, are deterministic algorithms that derive pseudo random numbers by inputting seeds. However, Magniber v2's RNG is developed by an attacker and its security is not guaranteed. We reveal the inherent vulnerabilities in Magniber v2's RNG and exploit them to recover the key parameters such as encryption key and initialization vector (IV). The vulnerability is derived from the seed input and structure of the RNG.
3.  Once the encryption key has been recovered, its correctness must be verified. Valid encryption key verification can apply a fixed known value, commonly called an authenticator. However, an authentication based verification is difficult when the ransomware encrypts various files. To solve this problem, we apply padding verification and statistical randomness tests. Padding verification is computationally cheaper than randomness tests, but gives a high false-positive rate. To compensate this deficiency, we perform additional randomness tests on the objects passed by the padding verification. By reducing the number of randomness tests, we improve the efficiency of the encryption key verification.

## 2. Related Work

There are various techniques and tools developed for detecting and decryption different types of ransomware. Ransomware detection systems determine whether it is ransomware through static or dynamic analysis. The static analysis-based detection identifies ransomware by using information such as strings, Application Programming Interface (API), signatures and metadata in the ransomware binary. Kanwal, Meet et al developed a model that can statically detect ransomware in the Android system by collecting strings that frequently used in ransom notes such as "_RANSOM", "_locked", and "_money" and analyzing API [8]. Similarly, Andronio et al proposed a model that can detect ransomware in Android systems by comprehensively analyzing API related to data encryption and device locking [9]. Since static analysis-based detection analyzes binary files, it is possible to diagnose before ransomware execution, which is a technique often used by antivirus. However, it is difficult to apply the technique if the ransomware is packed or code obfuscated [10]. The dynamic analysis-based detection collects and detects behavior data generated when ransomware is running. The behavior data includes information such as

entropy, I/O frequency, API, etc. Unlike static analysis, the method has a high detection rate as the behavior of the actual ransomware does not change even if obfuscation or packing is applied. It also has a high detection rate in new variants. However, some files are lost because the actual ransomware needs to be executed, and if false-positive occurs, all files are at risk of being encrypted [11–15].

Recently, various ransomware detection studies using artificial intelligence techniques have researched. Hasan, Md Mahbub et al developed a ransomware detection framework. The Author extracted more than 60 features including API Call, registry, file system and process operation, and trained using Support Vector Machine (SVM) algorithm [16]. Lee et al. used machine learning algorithm to detect if files were encrypted before being copied to the backup file system. Each file's entropy was estimated using some of the NIST900-80b tests, and it was trained via the model of k-NN, Decision Tree, kernel support vector machine and Multi-Layer Perceptrons algorithms [17]. Almashhadani et al. described a detection system that extracted the features exclusively from network traffic. The authors obtained twenty features from the characteristics of TCP, HTTP and DNS traffic. They made a decision making module based on two classifiers that one classifier was based on per-packet features, the other was based on flow-based features. The classifiers evaluated were Random Forests, Bayes Networks, SVM and Random Trees [18].

Decryption studies fall into two main categories: targeting multiple ransomware and individual ransomware. Targeting multiple ransomware focuses on restoring the random number used in the current encryption key by hooking or monitoring the PRNG function. Kolodenker et al. focused on the fact that ransomware employing hybrid cryptosystem creates symmetric-keys during infected [19]. Symmetric keys are encrypted and stored by the attacker's public key and can be decrypted by a private key held by the attacker. As the private key is difficult to acquire, the authors suggested setting up a key escrow, observing the symmetric keys used for file encryptions and hooking the Crypto API in Windows to acquire the session keys. Similarly, Kim et al proposed that the cryptographically secure pseudo random number generator (CSPRNG, provided by Windows as CryptoGenRandom) can be replaced by a user-defined deterministic random bit generator (DRBG) [20]. They suggested storing the seed used in the DRBG in an external repository and reproducing the encryption key using stored seed when a ransomware infection occurs.

Meanwhile, individual ransomware is decrypted by guessing the key (exploiting, if present, the cryptographic vulnerability of a ransomware), or extracting the key from memory (exploiting the system vulnerability). Hidden-Tear is an open source ransomware released in August 2015. Originally created for educational purposes, it has been modified by cybercriminals to spread various ransomware. Hidden-Tear uses the AES algorithm for file encryption and sends the key to the C&C server [21]. However, a vulnerability in the Hidden-Tear design enables the decryption of infected files. In particular, the PRNG seed of Hidden-Tear is predictable and uses a hard-coded IV and salt data. The author has published the Hidden-Tear decryption tool, meaning that variants with the same PRNG can be decrypted [22]. The PRNG of Donut ransomware, a variant of Hidden-Tear that appeared in 2018, differs from that of Hidden-Tear, so cannot be decrypted with the Hidden-Tear decryption tool. Lee et al. found that because the key of Donut is not deleted from memory, it can be extracted from memory and used for file decryption [23]. They also proposed a method that decrypts LooCipher, a ransomware that appeared in 2019 [24] and encrypts all files with AES128-ECB using a unique key. They found that the LooCipher's PRNG uses a $Random()$ function that is seeded with the current time, so the unique key can be recreated using the time of infection.

WannaCry ransomware that has infected more than 200,000 computers in 150 countries around the world since May of 2017 [25]. This ransomware encrypts files using AES and encrypts an AES key with a locally generated RSA implemented through a Crypto API [26]. However, in some versions of Windows XP and Windows 7, the prime number of the RSA private key of the Crypto API is not deleted from memory. By exploiting these vulnerabilities, the decryption tool Wanakiwi was developed to extract it existing

in memory to recovering the RSA private key [27]. Boczan reported the evolutionary changes in the cryptographic algorithms of GandCrab ransomware (versions 1–5) and the packing technique of GandCrab that bypasses antivirus software [28]. In GandCrab versions 1, 2 and 3, the file encryption key and IV are generated by a CSPRNG, and the file is encrypted using an AES. The AES key and IV are then encrypted with a locally generated RSA public key and the RSA key pair is transmitted to the C&C server. In versions 4 and 5, the AES is replaced by the stream cipher Salsa20 for faster encryption, the encryption system newly encrypts the existing locally generated RSA private key with the attacker's public key to ensure that encryption works well even when communication is impossible. Currently, the GandCrab v1, v4 and v5.0–v5.2 only are known to be decryptable. To prevent the damage by GandCrab, the Federal Bureau of Investigation, several law enforcement agencies, and individual companies have collaborated to obtain a master private key [29,30]. In addition, European law enforcement and IT Security companies have launched the No More Ransomware project to mitigate the impact of ransomware on business and individuals [31]. This collaborative project offers decryption tools for a range of ransomware.

## 3. Analysis of Magniber v2

This section details the encryption process of Magniber v2 together with its malicious behavior process, which we reveal by reverse engineering.

### 3.1. Malicious Behavior Process

Figure 1 shows the 10 steps of the entire malicious behavior process. Magniber v2 begins by running Loader.dll, which is installed and executed on the victim's PC through the malvertising technique. Loader.dll unpacks Payload.dll (1), which performs the malicious actions, searches from the target process (2) and injects dll into process (3). After identifying the system language, Payload.dll performs malicious actions on victim's computers (4).

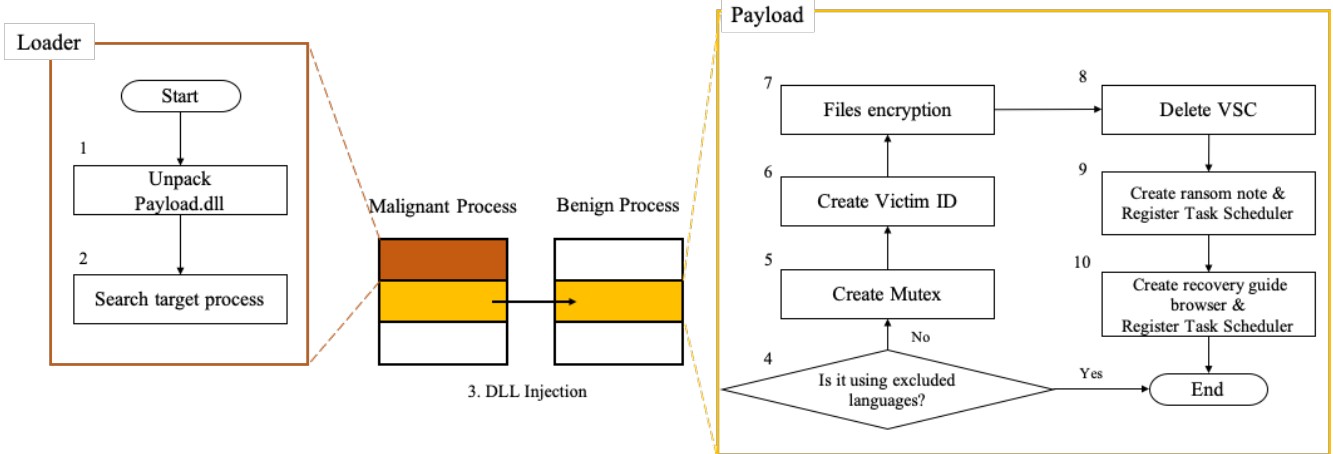

**Figure 1.** Entire process of malicious behavior by Magniber v2.

Thereafter, Payload.dll generates a mutex (5) that encrypts the files in a non-duplicable way. For victim identification purposes, its PRNG generates a victim's ID (6). The PRNG is customized directly by the developer of Magniber v2 and generates both the IV and encryption key. The structure of the Magniber v2 PRNG, which plays a major role in our research, is detailed in the next section. The files to be encrypted are documents, media, ,image, audio and etc., with 748 extensions in total(7). Folders required for program files, booting and running the operation system are excluded from encryption. Finally, the ransomware deletes the Volume Shadow Copy (VSC) would otherwise allow a system restore (8) and registers the ransom note and recovery guide browser on the task scheduler. The ransomware then runs automatically at the designated time (9, 10).

A working sample of Magniber v2 has been obtained in a hybrid analysis [32]. The hash value of each object is as follows.

1.  Loader.dll
    - MD5: 72FCE87A976667A8C09ED844564ADC75
    - SHA1: D3E17F5ECA5FB23B272549692D84CC449CF71527
    - SHA256: 6E57159209611F2531104449F4BB86A7621FB9FBC2E90ADD2ECDFBE 293AA9DFC
2.  Payload.dll
    - MD5: 19599CAD1BBCA18AC6473E64710443B7
    - SHA1: F9E2111E2903838BB9F4EFB557F75745D028BC3E
    - SHA256: FB6C80AE783C1881487F2376F5CACE7532C5EADFC170B39E06E174 92652581C2

### 3.2. Encryption Process

Because our aim is to decrypt infected files, we first describe the encryption process in detail. The order of file infection by Magniber v2 is determined by Depth First Search (DFS) (Depth-first search is an algorithm that traverses or searches tree or graph data structures. The algorithm starts at the root node (on a graph, the root is an arbitrary node) and explores as far as possible along each branch before backtracking).

Folders at the same depth are searched and infected in ascending order of their folder names. Next, files in the same folder are preferentially infected in ascending order of their file names. File encryption by Magniber v2 proceeds in three steps: file encryption key (*FEK*) and *IV* generation, *FEK* and *IV* encryption and file encryption. These steps are applied to each target file. Each step of the Magniber v2 file encryption process is presented in Figure 2 and described below.

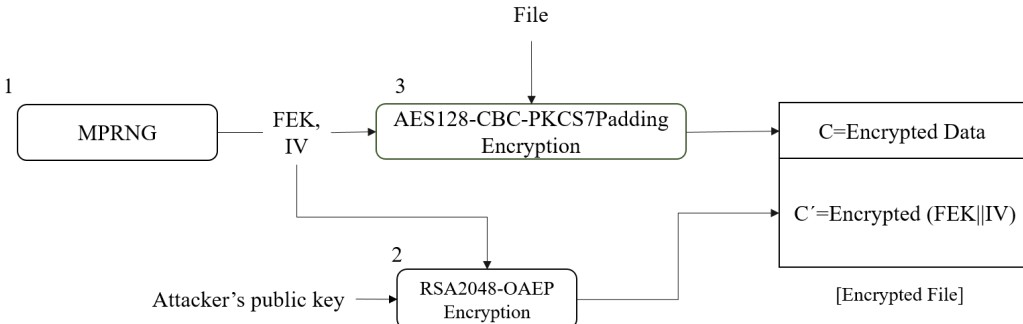

**Figure 2.** Encryption process of Magniber v2.

### 3.2.1. *FEK* and *IV* Generation

Using its PRNG, Magniber v2 generates different 16-byte *FEK* and 16-byte *IV* for file encryption. This PRNG (hereinafter referred as to the Magniber PRNG (MPRNG)) is not a well-known algorithm such as Hash DRBG, HMAC DRBG and CTR DRBG, but is custom-made by the attacker. MPRNG is seeded by the output of GetTickCount and outputs a random string composed of ASCII characters from '0' to '9' and 'a' to 'z'. As MPRNG internally calls the GetTickCount function twice while generating each one-byte random character, it calls the GetTickCount function $2n$ times when deriving an $n$-byte random string. The structure of MPRNG is important for decrypting an encrypted file, as will be explained in detail in the next section.

### 3.2.2. *FEK* and *IV* Encryption

Magniber v2 encrypts the key parameters of file encryption, *FEK* and *IV*, using the hard-coded public key developed by the attacker. The use of a public (hard-coded) key for encryption, which is the inability to decrypt the ciphertext (encrypted key parameters)

without the private key, are the main features of asymmetric (public-key) ciphers. The *FEK* and *IV* are concatenated and encrypted with RSA2048-OAEP as shown in Equation (1).

$$C' = \text{RSA2048-OAEP}(FEK||IV, PubKey) \tag{1}$$

where $C'$ is the encrypted value of the concatenated *FEK* and *IV*, *PubKey* is the attacker's RSA public key, which is hard-coded in Magniber v2.

### 3.2.3. File Encryption

File encryption is performed by the encryption algorithm AES128 using the Cipher Block Chaining (CBC) mode of operation and PKCS#7 padding (denoted AES128-CBC-PKCS7Padding). This step is described by

$$C = \text{AES128-CBC-PKCS7Padding}(P, FEK, IV) \tag{2}$$

Here, *P* is a file to be encrypted and *C* is an encrypted file. The encrypted file is created by concatenating *C* to *C'* (*C* and *C'* are outputs of Equations (1) and (2), respectively). Once the file encryption is completed, the *FEK* and *IV* are then zeroized, which prevents their recovery through memory analysis.

## 4. Method for Finding the Encryption Keys of Magniber v2

In general, ransomware employing hybrid encryption can maintain its malicious behavior provided that the the attacker's private key is unknown. Although Magniber v2 uses hybrid encryption, we have found a technique that recover the *FEK* and *IV* of the file encryption without requiring the attacker's RSA private key, resulting in the success of file decryption.

### 4.1. Magniber PRNG

The MPRNG derives the *FEK* and *IV* used in file encryption. The Magniber v2 RNG function is given in Algorithm 1. The inputs *m* and *M* determine the range of the output *r*. The global parameter *feedback* retains the stored value until the end of the process. The RNG, which includes a Feedback Calculator (FC) and a filter, outputs a one-byte *r* between *m* and *M*. The FC receives (as inputs) the previous *feedback* and the output of the GetTickCount function that returns the current *tick* used as a seed, and outputs a four-byte *feedback*. The current *tick* returned by the GetTickCount function is the number of milliseconds that have elapsed since the system was started. The *feedback* passes through the Filter and outputs 1-byte *r* between *m* and *M*. When the RNG is first called, the previous *feedback* does not exist, so Steps 1 and 2 instead call the GetTickCount function, which provides the *tick* input. In Step 9, the *feedback* uses only 20 bits due to masking with 0x11ee0fff. This vulnerability reduces the attack complexity of acquiring *feedback*, which can accelerate the *FEK* recovery, from $2^{32}$ to $2^{20}$. MPRNG is composed of concatenated RNGs as shown in Figure 3, and it is called repeatedly to output a random string composed of '0' to '9' and 'a' to 'z' characters. MPRNG inputs the desired length *n* and outputs an *n*-byte *R*. MPRNG generates a one-byte random character through two concatenated RNGs, and repeats this process *n* times to derive an *n*-byte random string. In the one-byte random character generation, the input parameters *m* and *M* of the first RNG are fixed at 0 and 1, respectively.

The output of the first RNG (0 or 1) determines the input of the second RNG: if the output is 0, the second RNG's input is 'a' and 'z'; otherwise, its input is '0' and '9'. The second RNG then outputs a one-byte character between 'a' to 'z' or '0' to '9'. Repeating this process *n* times generates an *n*-byte random string. The MPRNG algorithm is given in Algorithm 2.



---

**Algorithm 1** RNG of Magniber v2

---

**Function** RNG($m, M$)

**Input:** Minimum $m$ and maximum $M$ values

**Output:** 1-byte random character $r$ between $m$ and $M$

**Global:** *feedback*

     // *Initialization* is performed only on the initial call.

1: **if** Initialization **then**

2:     $G \leftarrow$ GetTickCount()

3: **else**

4:     $G \leftarrow feedback$

5: **end if**

     // subfunction **Feedback Calculator**

6: $feedback \leftarrow G \oplus 1$

7: $tick \leftarrow$ GetTickCount()

8: $feedback \leftarrow (tick + feedback + (tick + feedback)$

9: $feedback \leftarrow (feedback \mod 0x3e8)\&0x11ee0fff$

     // subfunction **Filter**

10: $r \leftarrow m + feedback \mod (M - m + 1)$

11: **return** $r$

---

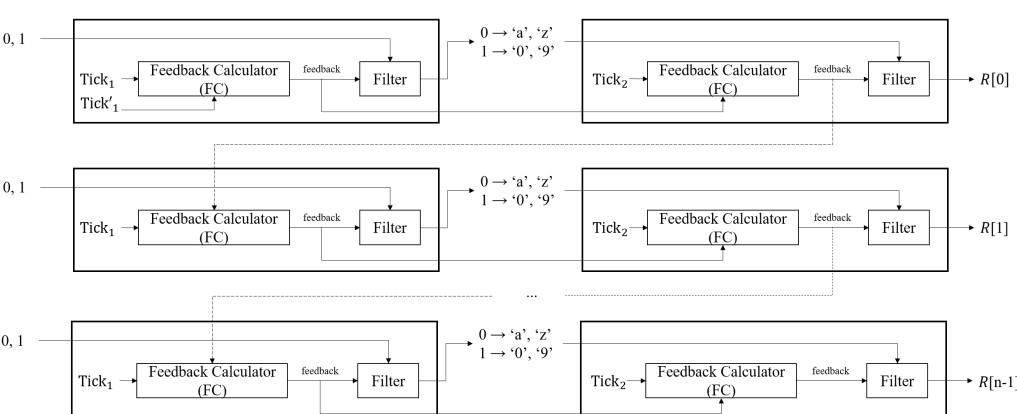

**Figure 3.** Structure of Magniber's Pseudo Random Number Generator (PRNG), Magniber PRNG (MPRNG).

---

**Algorithm 2** PRNG of Magniber v2

---

**Function** MPRNG($n$)

**Input:** Desired random string length $n$

**Output:** $n$-byte random string $R$

1: **for** $i \leftarrow 0$ **to** $n - 1$ **do**

2:     $m \leftarrow 0$, $M \leftarrow 1$

3:     **if** RNG($m, M$) is zero **then**

4:         $m \leftarrow$ 'a', $M \leftarrow$ 'z'

5:     **else**

6:         $m \leftarrow$ '0', $M \leftarrow$ '9'

7:     **end if**

8:     $R_i \leftarrow$ RNG($m, M$)

9: **end for**

10: **return** $R$

---

The MPRNG derives the victim's ID, *FEK* and *IV*. To acquire the current *tick* used as a seed, it calls the GetTickCount function $2n$ times and generates an $n$-byte random string. Therefore, when attempting to recover an $n$-byte random string, we must acquire the $2n$ *tick*s in order of their generation, which is an apparently intractable task. However, we have observed that the number of *tick* changes is zero or very small. That is, an $n$-byte random string can be recovered after significantly fewer guesses than intended by the MPRNG developer. The security of MPRNG is guaranteed by the multiple *tick*s used as seeds and the four-byte *feedback* that continuously affects the next RNG output. However, we found defects in two characteristics of MPRNG, which were exploited for *FEK* and *IV* recovery as described in Section 4.2.

### 4.2. Generating File Encryption Key Candidates

Although we reverse-engineered the entire encryption process of Magniber v2, we could not decrypt the infected file without acquiring the attacker's RSA private key or *FEK* and *IV* of each file. As the RSA private key is owned only by the attacker, we focused on obtaining the *FEK* and *IV* of encryption infection.

The *FEK* candidates were generated by guessing their *tick* and *feedback* values. First, we found the *tick*s for deriving the *FEK*. Figure 4 shows four cases of all possible patterns of *tick* changes when generating an $n$-byte random string. Note that the same number of *tick*s can be generated with different timings.

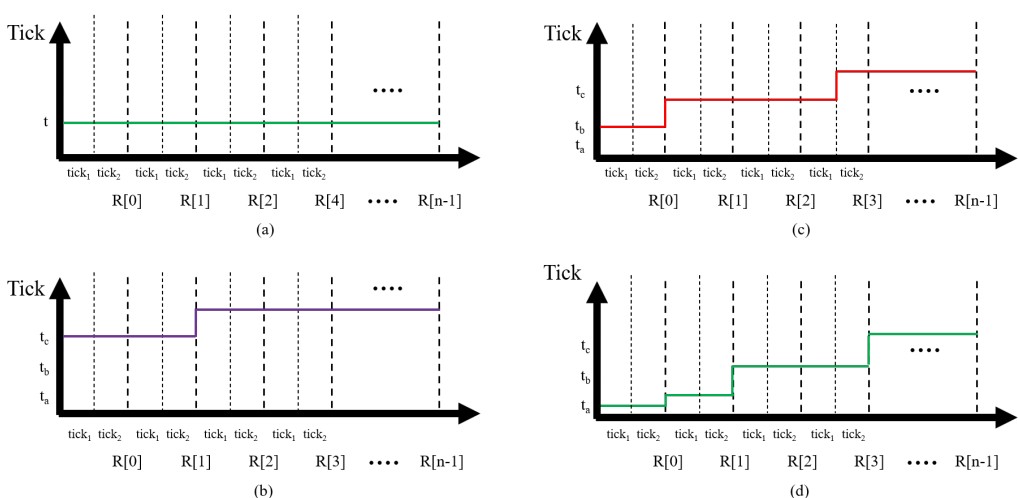

**Figure 4.** Tick changes during generation of an $n$-byte random string: (**a**) no tick changed, (**b**) one-tick changed, (**c**) two-tick changed and (**d**) three-tick changed

For example, in Figure 4a, an $n$-byte random string is generated with the same *tick*. However, if there is one *tick* change at the time-point of calling $tick_2$ of R[$l$] ($0 < l < n - 1$) (where R[$l$] denotes the $l$ th byte of R), that *tick* endures from $tick_1$ of R[0] to $tick_1$ of R[$l$], and also from $tick_2$ of R[$l$] to $tick_2$ of R[$n - 1$]. Therefore, we must guess the values of the *tick*s not only from the number of *tick* changes, but also from the time-points of their changes. We next found the four-byte *feedback*. The *feedback* is the value by which the current call affects the next call in the MPRNG. Therefore, obtaining the *feedback* injected into the first byte of $FEK_i$ ($0 \leq i < n$) is essential for a full $FEK_i$ recovery, where $FEK_i$ is the encryption key of File$_i$. More specifically, we organized the relationship of the parameters inserted to the MPRNG in chronological order. After injection, MPRNG is first called for generating the victim's ID. Thereafter, it is repeatedly called in the sequential derivation of $FEK_i$ and $IV_i$ for the encryption of each File$_i$. As shown in Figure 5, the *feedback* after the victim's ID generation is injected into the $FEK_0$ derivation process, and the *feedback* after the $IV_{i-1}$ generation is injected into the $FEK_i$ ($i \geq 1$) derivation.

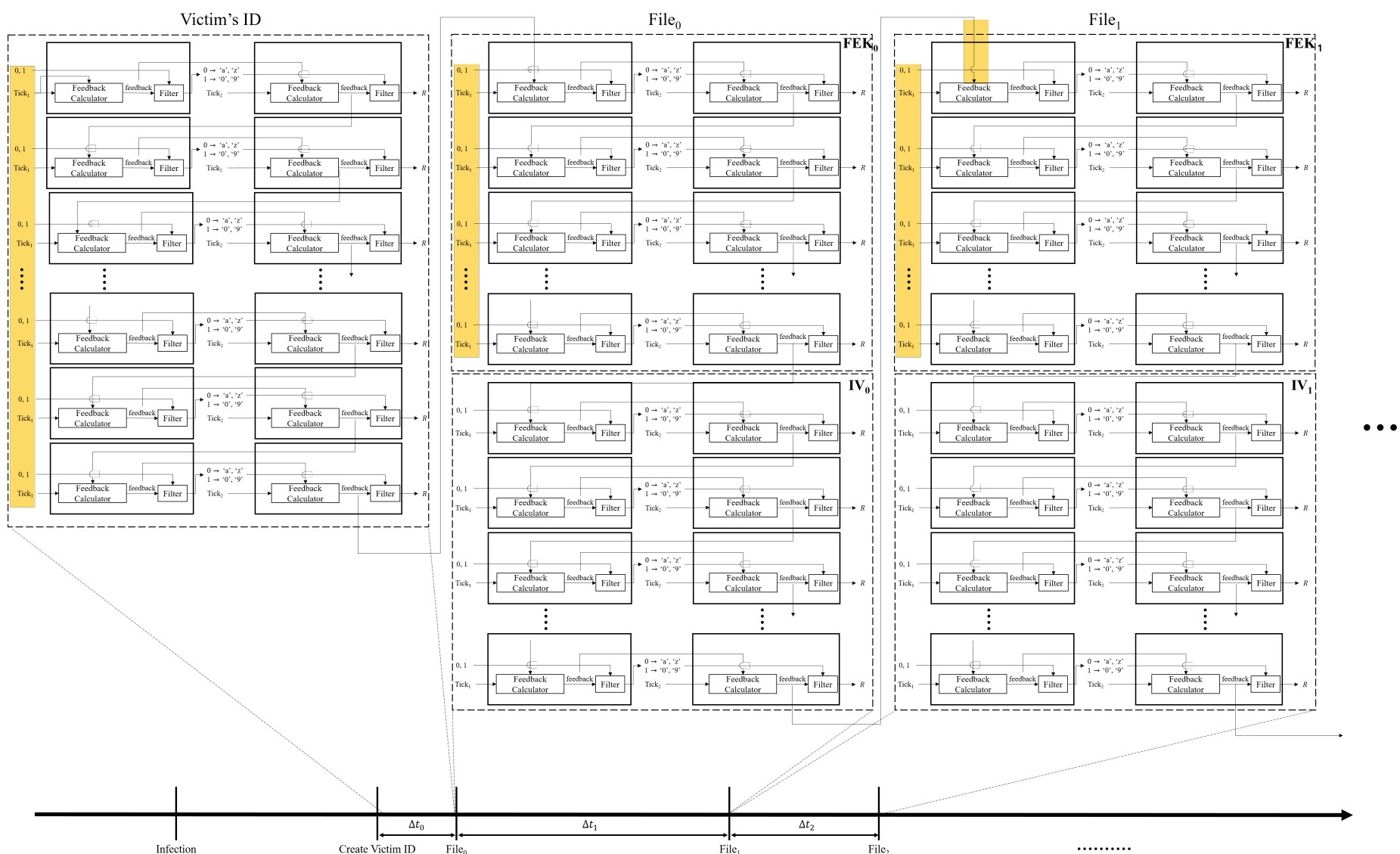

**Figure 5.** Entire process of MPRNG during infection.

The *feedback* injected into the $FEK_0[0]$ derivation is generated after deriving the last byte of the victim's ID. Using the known victim's ID included in the ransom note, which is known (see Figure 6), the *feedback*s can be guessed by estimating the *tick*s alone.

As the MPRNG is initialized during the victim's ID generation, we need only guess the *tick*s; a separate *feedback* guess is not required. If we derive the correct victim's ID, we automatically get the *feedback* injected into the $FEK_0$ derivation. The $FEK_i$ is generated after File$_1$ is injected with the *feedback* generated after the $IV_{i-1}$ derivation of File$_{i-1}$. In our decryption process, finding the *tick* used to generate the victim's ID is important for another reason: the GetTickCount function returns the time elapsed (in milliseconds) since the system is started. However, as the system startup times differ among the users, we require a reference point for the *tick* inference. Here, we set the *tick* used in creating the victim's ID as the reference point.

```
ALL YOUR DOCUMENTS PHOTOS DATABASES AND OTHER IMPORTANT FILES HAVE BEEN ENCRYPTED!
====================================================================================
Your files are NOT damaged! Your files are modified only. This modification is reversible.

The only 1 way to decrypt your files is to receive the private key and decryption program.

Any attempts to restore your files with the third party software will be fatal for your files!
====================================================================================
To receive the private key and decryption program follow the instructions below:

1. Download "Tor Browser" from https://www.torproject.org/ and install it.

2. In the "Tor Browser" open your personal page here:

http://ws8exfvv4xxm477hv3a.wdjiqa44qtkw7uqy.onion/dyaaghemy
```

**Figure 6.** Ransom note of Magniber v2 (the victim's ID is enclosed in the red-edged box).

Once established, the reference provides a starting point for sequentially guessing the *tick*s from the order of infection. If the *tick* changes during $IV_{i-1}$ generation, the $IV_{i-1}$ can be verified only when the first block of File$_{i-1}$ is fixed and known (unlike the case of the victim's ID). However, the first block of the file is rarely fixed. Even for files of known format, such as doc, hwp and pdf files, the first block is a variable header area of limited use in verification. Despite these limitations, the *feedback* injected into the $FEK_i$ derivation can be acquired by brute-force attack of the $2^{20}$ possibilities without recovering the $IV_{i-1}$, but merely by exploiting the vulnerability of MPRNG described in Section 4.1. In the first block that is not decrypted, file operation is possible only by filling an appropriate value in a corresponding part in a known file format. The $FEK_{i-1}$ recovery naturally acquires the *tick* used for generating the $FEK_{i-1}[15]$. This *tick* is continuously injected into the $IV_{i-1}$ generation, and based on whether it changes or not, we can know the *feedback* injected with $FEK_i[0]$. The acquisition of the *feedback* greatly affects the efficiency of our decrypting performance. If the *feedback* is obtained, the attack burden of $2^{20}$ possibilities can be reduced. We therefore recover the $FEK_i$ assuming no *tick* change in the $IV_{i-1}$ generation. If the recovery fails, we perform the brute-force attack based on $2^{20}$ *feedback* candidates. The recovery algorithm adopting this scheme is developed in the next section.

Algorithm 3 generates a list $L$ of $FEK_i$ candidates with one *tick* change. The $L$ is compiled according to the position and interval of the changed *tick* relative to the reference point (the *base*). An algorithm based on Algorithm 3 can also generate a list of $FEK_i$ candidates with more than one *tick* change.

In Algorithm 3, *pos* represents the position of the *tick* change, and the time interval ranges from the starting point (*base*) to the end point (*endtick*). Steps 4 and 5 generate $FEK$ candidates within the time range. We introduce a *tick* change in the MPRNG by modifying the argument with two additional *tick* arguments, $tick_1$ and $tick_2$ for mPRNG($n$,$tick_1$,$tick_2$), which are input to the first and second RNG, respectively. The *tick* input to the mPRNG is divided into *tick* before the change and *duration* after the change. In Step 4, $candidateFEK_1$ is mPRNG($pos-1$, *tick*, *tick*)$\|$ mPRNG(1, *duration*, *duration*)$\|$ mPRNG(16-$pos$, *duration*, *duration*), indicating a *tick* change to $tick_1$ of *pos* position mPRNG. Similarly, in Step 5, a *tick* change occurred at $tick_2$ of the *pos* position mPRNG.

---

**Algorithm 3** $FEK_i$ Candidates Generation Algorithm

---

**Function** DeriveFEKCandidates($base, \Delta t$)
**Input:** Reference point $base$, Time interval $\Delta t$
**Output:** List of FEK Candidates $L$

 1:  **for** $pos \leftarrow 1$ **to** 16 **do**
 2:     **for** $tick \leftarrow base$ **to** $endtick \leftarrow base + \Delta t$ **do**
 3:        **for** $duration \leftarrow tick$ **to** $endtick$ **do**
 4:          // A *tick* change $tick_1$ occurs at *pos* position mPRNG.
 5:          $candidateFEK_1 \leftarrow$ mPRNG($pos - 1, tick, tick$)$\|$ mPRNG($1, duration, duration$)$\|$
                          mPRNG($16 - pos, duration, duration$)
 6:          // A *tick* change $tick_2$ occurs at *pos* position mPRNG.
 7:          $candidateFEK_2 \leftarrow$ mPRNG($pos - 1, tick, tick$)$\|$mPRNG($1, tick, duration$)$\|$
                          mPRNG($16 - pos, duration, duration$)
          // Generate *IV* without changing *tick*
 8:          $IV \leftarrow$ mPRNG($16, duration, duration$)
 9:          Include $candidateFEK_1 \| IV$ and $candidateFEK_2 \| IV$ in $L$
10:        **end for**
11:     **end for**
12:  **end for**
13:  **return** $L$

---

## 5. Recovery of Encryption Keys and Decryption of Files Encrypted by Magniber v2

### 5.1. Verifying the Generated File Encryption Key Candidates

Now, we are ready to recover the $FEK_i$. The $FEK_i$ recovery process is performed in two steps: padding verification and statistical randomness tests to verify the correctness of the $FEK_i$ candidates. We considered the point that it is possible to decrypt all the encrypted blocks except the first block without the $IV_i$ as the error propagation property of the CBC mode of operation used for file encryption. That is, if the guessed $FEK_i$ candidate is correct, the padding verification accepts $File_i$, which is the decryption result by $FEK_i$. However, in some cases, the padding verification yields a false positive for an incorrect $FEK_i$ candidate. To remove these false candidates, we checked whether $File_i$ was correctly decrypted in statistical randomness tests. Although there tests reduce the false-positive rate of detecting $FEK_i$ candidates, they incur much higher computational cost than padding verification.

Padding verification of decrypted files: The padding, which matches the length of the plaintext before encryption to a multiple of the block length of the symmetric-key cipher, occupies the last block. The ciphertext is obtained by encrypting the padded plaintext. Conversely, the plaintext is obtained by decrypting the ciphertext and removing the padding. Before removing the padding from the decrypted ciphertext, the correctness of the padding must be verified. If the padding fails the verification test, then the decryption of the ciphertext is incorrect. This mechanism can identify the correct $FEK_i$s among the $FEK_i$ candidates. We decrypt the data $C_i$ of $File_i$ encrypted with $FEK_i$, and then verify the padding in the last block of decrypted $C_i$. In the padding verification, we decrypt only the last block (not the entire encrypted file) for efficiency. The padding value of PKCS7Padding used by Magniber v2 is determined by the padding length: for instance, a one-byte padding length is padded with 0x01, whereas a four-byte padding length is padded with 0x04040404. Therefore, we check the value from the last byte of the last block and verify the correctness of the padding with that value. Padding verification is relatively inexpensive because it performs a simple value-checking operation. However, we identified the occurrence of false-positive $FEK_i$ candidates in the padding verification. The false positive probability depends on the validated padding length. For example, if a padding value of 0x030303 passes the verification test, its probability of being a false positive is about $256^{-3}$. More generally, the false-positive probability of a length-$n$ padding is about $256^{-n}$. Clearly, the

false-positive rate of the padding verification increases with decreasing padding length. Therefore, files that passed the padding verification were re-tested more rigorously to find the correct $FEK_i$ candidates.

Randomness tests for decrypted files: Our verification method acknowledges that plaintext can be recovered only by decrypting the infected file with the correct $FEK_i$. In other words, we must distinguish whether an infected file decrypted with a $FEK_i$ candidate is plaintext or ciphertext (an infected file decrypted with an incorrect $FEK_i$ is also expressed as ciphertext). To distinguish between plaintext and ciphertext, we considered the randomness feature of ciphertext and apply the statistical randomness suite NIST SP800-22, which consists of 15 randomness tests. The randomness tests were performed on the data of 600 files (100 files in each of the following formats: video, audio, image, document, compressed and encrypted). The NIST recommends 25 runs of each test to ensure randomness. However, one run of each test is sufficient for our purposes. The statistical randomness results of our test set are listed in Table 2.

**Table 2.** Results of statistical randomness tests on our test set.

| Test List (ms/Test *) | Video | Image | Audio | Document | Compressed | Encrypted | False Positive Rate |
|---|---|---|---|---|---|---|---|
| Frequency (4) | 2 | 8 | 11 | 5 | 5 | 100 | 5.17% |
| Block frequency (2) | 0 | 0 | 0 | 0 | 3 | 98 | 0.83% |
| Runs (4) | 0 | 1 | 1 | 1 | 5 | 97 | 1.83% |
| Longest run (5) | 33 | 0 | 4 | 10 | 31 | 100 | 13% |
| Rank (47) | 8 | 7 | 50 | 33 | 55 | 99 | 25.67% |
| FFT (188) | 4 | 2 | 32 | 12 | 41 | 98 | 15.5% |
| Non overlapping template (1167) | 0 | 0 | 0 | 1 | 4 | 100 | 0.83% |
| Overlapping template (35) | 28 | 8 | 4 | 1 | 24 | 100 | 10.83% |
| Universal (30) | 0 | 0 | 0 | 0 | 8 | 96 | 2% |
| Approximate entropy (124) | 0 | 0 | 0 | 0 | 4 | 100 | 0.67% |
| Serial (343) | 0 | 0 | 0 | 0 | 8 | 97 | 1.83% |
| Linear complexity (1890) | 17 | 10 | 47 | 23 | 56 | 100 | 25.5% |
| Cumulative sums (5) | 1 | 0 | 0 | 1 | 4 | 100 | 1% |
| Random excursions (4) | 0 | 0 | 0 | 0 | 2 | 65 | 6.17% |
| Random excursions variants (4) | 0 | 0 | 1 | 0 | 2 | 62 | 6.83% |

These tests are the results for *p*-value = 0.01. * Time per test in milliseconds.

Based on the experimental results shown in Table 2, we selected four randomness tests (block frequency, runs, universal, cumulative sums) as suitable tests of $FEK_i$ recovery. In selecting these tests, we considered not only the distinction rate between plaintext and ciphertext, but also the test speed. The non-overlapping template, serial and approximate entropy tests gave excellent distinction rates between plaintext and ciphertext, but were slower than the other selected tests. Conversely, the frequency, longest run, random excursions and random excursions variants ran quickly but yielded low distinction rates.

We applied four selected tests to the $FEK_i$, which passed the padding verification. Any $FEK_i$ candidate that passed the padding verification, but not passed randomness tests was concluded as the correct $FEK_i$.

### 5.2. File Encryption Key Recovery

Algorithm 4 gives our $FEK_i$ recovery algorithm developed through Section 4.2 and 5.1. This algorithm inputs three arguments ($File_i$, *base* and $\Delta t$) and outputs the correct $FEK_i$. Here, $File_i$ is the *i*-th infected file to be decrypted. In the $FEK_i$ recovery process, the recovery order of $FEK_i$ is important because the previous value of the MPRNG affects the next value. The $FEK_i$s are recovered in the order of their file infection in Magniber v2 based on DFS. The base value provides the reference point of each $FEK_i$ recovery. The initial *base* in $FEK_0$ recovery is the *tick* used to generate the last byte of the victim's ID. In subsequent recoveries, *base* is the *tick* used to generate $FEK_{i-1}$[15]. $\Delta t$ represents the time-change range of the *tick*.

This value is arbitrary but must be appropriately set to balance its trade-off relationship with recovery time. Algorithm 4 is divided into two parts, depending on whether or not the *tick* changes during the $IV_i$ generation. Steps 1–10 are executed when the *tick* is unchanged in the $IV_{i-1}$ derivation. Step 1 calls Algorithm 3, which generates the list $L$ of $FEK_i$ candidates. Here, *feedback* applies the value generated after deriving $FEK_{i-1}$[15]. Steps 3–9 are then executed on the $FEK_i$ candidates in $L$. First, Step 3 decrypts $File_i$ with AES128-CBC using candidate $FEK$ and $IV$, and obtains plaintext $P$. Step 4 performs the padding verification of $P$, and Step 5 performs the four randomness tests if $P$ passes the padding verification; that is, if candidate $FEK$ corresponding to $P$ is concluded as the correct $FEK_i$. If the $FEK_i$ recovery fails in Steps 1–9, the algorithm performs a $2^{20}$ *feedback* brute-force attack (Steps 11–24). Apart from the brute-force *feedback*, this process operates similarly to the previous process. RNG, a subfunction of MPRNG, is masked with 0x11ee0fff, so the effective number of bits is reduced from 32 to 20. Steps 12 and 13 exploit this vulnerability in the *feedback* reconstruction. After constructing the $FEK_i$ candidate list $L$ based on the reconstructed *feedback*, the process executes identically to the previous process.

---

**Algorithm 4** $FEK_i$ Recovery Algorithm

---

**Function** recoveryFEK ($File_i$, $base$, $\Delta t$)
**Input:** Encrypted file data $File_i$, Reference point $base$, Time interval $\Delta t$
**Output:** Correct $FEK_i$
**Global** *feedback*
　　// Assume that the *tick* was not changed in the $IV_{i-1}$ generation.
　1: $L$ = DeriveFEKCandidates($base$, $\Delta t$)
　2: **for** *CandidateFEK* in $L$ **do**
　3: 　　$P \leftarrow$ AES128-CBC($File_i$, $candidataFEK$, $IV$)
　4: 　　**if** isPadding($P$) is True **then**
　5: 　　　**if** *BlockFreqTest* and *RunsTest* and *CumSumTest* and *UniversalTest* is not pass **then**
　6: 　　　　$FEK_i \leftarrow candidataFEK$
　7: 　　　　**return** $FEK_i$
　8: 　　　**end if**
　9: 　　**end if**
　10: **end for**
　　// Assume that the *tick* was changed in the $IV_{i-1}$ generation.
　11: **for** $i \leftarrow 0$ to $2^{20} - 1$ **do**
　12: 　$feedback \leftarrow ((i\&0x80000) \ll 9) \| ((i\&0x78000) \ll 6)$
　13: 　$feedback \leftarrow feedback \| ((i\&0x7000) \ll 5) \| (i\&0xfff)$
　14: 　$L$ = DeriveFEKCandidates($base$, $\Delta t$)
　15: 　**for** *CandidateFEK* in $L$ **do**
　16: 　　$P \leftarrow$ AES128-CBC($File_i$, $candidataFEK$, $IV$)
　17: 　　**if** isPadding($P$) is True **then**
　18: 　　　**if** *BlockFreqTest* and *RunsTest* and *CumSumTest UniversalTest* is not pass **then**
　19: 　　　　$FEK_i \leftarrow candidataFEK$
　20: 　　　　**return** $FEK_i$
　21: 　　　**end if**
　22: 　　**end if**
　23: 　**end for**
　24: **end for**
　25: **return** *null*

---

### 5.3. Experimental Result

The key recovery speed was measured in the following system: 2 GB RAM, 2 processors and Windows 7 32-bit virtual environment. Within the virtual environment, common files were infected with Magniber v2, and the *FEK*s of the infected files were recovered by running Algorithm 4. During the experiment, the modification time interval of the encrypted files is not exceeded 50 ms. Therefore, we heuristically set $\Delta t$ to 50 ms.

Table 3 shows the experimental results. When the *tick* did not change during the $IV_{i-1}$ and $FEK_i$ generation, each $FEK_i$ was recovered in 0.140 s at most. When the *tick* changed once and twice during the $FEK_i$ derivation but no *tick* change occurred during $IV_{i-1}$, the recovery times per $FEK_i$ were 2.465 s and 1.5 min at most, respectively.

**Table 3.** Experimental results of file encryption key (*FEK*) recovery.

| Number of Tick Changes during $FEK_i$ | No Tick Changes during Generation of $IV_{i-1}$ | Tick Changes during Generation of $IV_{i-1}$ |
|:---:|:---:|:---:|
| 0 | 0.140 s | 56 min. |
| 1 | 2.465 s | 17 hrs. |
| 2 | 1.5 min. | 29 day. |

In these results, *t* was 50 ms.

Conversely, when the *tick* changed during the $IV_{i-1}$ derivation but not during the $FEK_i$ derivation, the maximum recovery time of including the brute force attack of *feedback* took 56 min, owing to the brute-force attack of *feedback*. In this circumstance, we estimated a one-*tick* and two-*tick* change in the $FEK_i$ derivation increased the recovery time to 14 h and 41 h per $FEK_i$, respectively.

Although a *FEK* with $2^{128}$ attack complexity cannot be found by normal approaches, the above results show that our decryption method can recover a *FEK* within practical time. Moreover, our experiment was performed on limited-resource hardware. The *FEK* recovery speed would improve if the infected files were moved into an environment with higher computing power, or if Algorithm 4 was parallelized during a feedback brute-force attack. Finally, we successfully decrypted the files using the recovered *FEK*s, as shown in Figure 7.

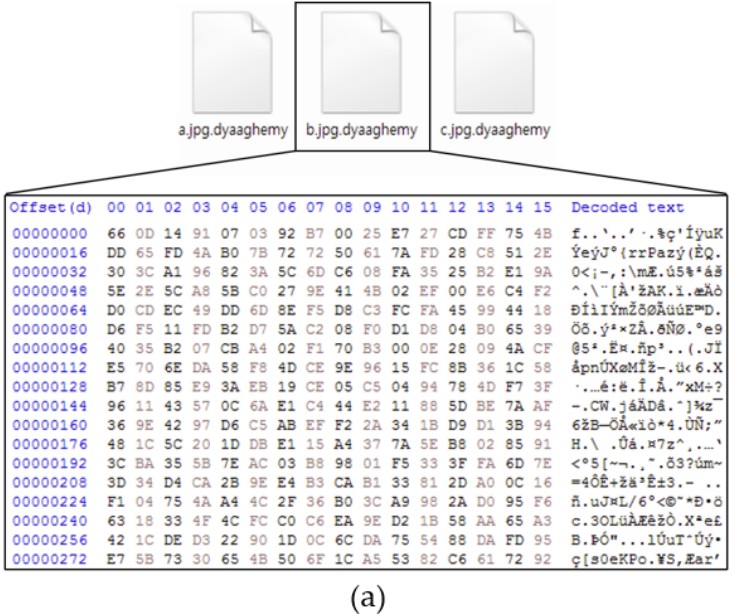

(a)

**Figure 7.** *Cont.*

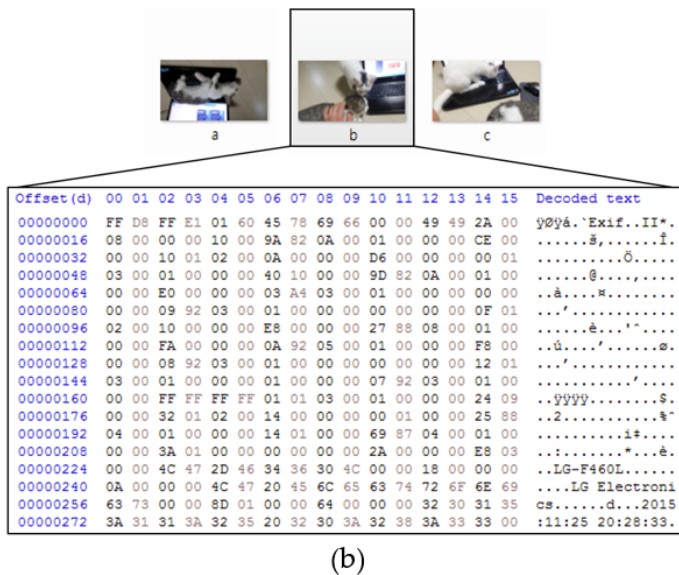

(b)

**Figure 7.** (**a**) Encrypted files (**b**) Decrypted files after applying our proposed method.

## 6. Conclusions

We analyzed Magniber v2 from a cryptographic viewpoint and studied file decryption through *FEK* recovery. We exploited a vulnerability in the MPRNG of Magniber v2 to generate the *FEK* and *IV*. In addition, we proposed an encryption key verification method that checks the existence of padding and performs statistical randomness tests. We demonstrated that the proposed method can decrypt Magniber v2-infected file without the knowledge of the attacker's private key.

Most of the latest ransomware use hybrid cryptographic algorithms. Files corrupted by these types of ransomware are commonly decrypted by acquiring the attacker's private key, which is used to encrypt the encryption keys of the corrupted files. However, if the PRNG that generates the file encryption keys is vulnerable, they can be retrieved by reproducing the seed at the time of the infection. We succeeded in decrypting infected files by recovering the file encryption key for Magniber v2, but files infected by ransomware using PRNG containing the same vulnerability can be decrypted using the proposed method. To complement this study, we will conduct a study on a system that can automatically detect vulnerabilities of PRNGs used in ransomwares.

**Author Contributions:** Formal Analysis, S.L.; Methodology M.P.; Writing—original draft, S.L.; Writing—reveiew & editing, M.P.; Project administration, J.K. All authors have read and agreed to the published version of the manuscript.

**Funding:** This work was supported by Institute for Information & communications Technology Promotion(IITP) grant funded by the Korea government(MSIT) (No. 2017-0-00520, Development of SCR-Friendly Symmetric Key Cryptosystem and Its Application Modes.

**Conflicts of Interest:** The authors declare no conflict of interest.

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
