# Peer review of "Magniber v2 Ransomware Decryption: Exploiting the Vulnerability of a Self-Developed Pseudo Random Number Generator"

_electronics, doi:10.3390/electronics10010016_

Round 1
Reviewer 1 Report
Dear Authors:
This is a very good paper that joins science and practice. The paper is easy to follow, very well-written and properly organized. Moreover, the presented methods were verified experimentally.
I strongly recommend acceptance of the manuscript, after resolving (very) minor issues:
- The following statement may cause misunderstandings (lines 37-39):
„One protection method is hybrid encryption1 File encryption is performed by a symmetric-key cipher: the attacker encrypts the encryption keys using its his/her public key and places the encrypted encryption keys in the victim’s system.”
To make it more understandable, I recommend moving the footnote to the main text.
- Please expand all important abbreviations (like “IV”, etc.).
- Please move Table 1 to the bottom of the page, since currently it appears before its reference in the main text.
- There are several strange linguistic phrases in the paper, such as “encrypted encryption”, please correct them.
- Line 149: “Fig. 1”->”Figure 1”.
Reviewer 2 Report
The authors analyzed Magniber v2 ransomware and its PRNG. They managed to successfully recover the encryption key by exploiting ransomware vulnerability.
This is well-done research and well-structured paper but nevertheless, there are some shortcomings that I suggest improving before accepting this paper.
1. in the title first part is not necessary (Magniber v2 Ransomware Decryption). Try to rewrite the title. It is only a suggestion (not obligatory).
2. Try to give some more information on the results in the abstract.
3. Rewrite Introduction so you give background, goal/purpose and contribution of your research. Related work can be in the new Section and there is no need to highlight that Related work analysis is in the domain of decryption of ransomware. It is understandable in itself.
4. Big question of this paper is its scientific soundness. You are solving one unique problem in this paper. Is this approach applicable to a whole class of problems, did you try the same approach to recover the encryption key on other ransomware? In this way, it is engineering, not science so highlight the broad applicability of this approach.
5. It is not clear what is planned in future research based on the conclusion taken from this research.
5. Check your references, you have a lot of internet sources that are not in reference style (date of access, etc.). Include scientific papers in respected journals (IEEE, Springer, ACM, Elsevier) especially in related work part. Literature has to be current.
Round 2
Reviewer 2 Report
Corrections that were made are sufficient. I am recommending this manuscript for publication.